# Perceived neighborhood social cohesion and functional disability among older adults: The moderating roles of sex, physical activity, and multi-morbidity

Kofi Awuviry-Newton[1,2]*, Dinah Amoah[1,3], Daniel Doh[1,4], Williams Agyemang-Duah[1,5], Kwadwo Ofori-Dua[1,6], Paul Kowal[7], Seth Christopher Yaw Appiah[6]

**1** African Health and Ageing Research Centre (AHaARC), Australia, **2** College of Science, Health and Engineering, Victoria University, Melbourne, Victoria, Australia, **3** School of Health Sciences, University of Tasmania, Australia, **4** School of Allied Health, University of Western Australia, Perth, Australia, **5** Department of Geography and Planning, Queen's University, Kingston, Ontario, Canada, **6** Department of Sociology and Social Work, Faculty of Social Sciences, Kwame Nkrumah University of Science and Technology, Kumasi, Ghana, **7** International Health Transitions, Canberra, Australia

* Kofi.Awuviry-Newton@vu.edu.au

**Data Availability Statement:** Minimal dataset has been included.

## Abstract

Though the Ghanaian social structure is largely communal in several of its social life and social spaces, the extent to which cohesive neighbourhood affects functional ability of older persons and the moderating factors of the relationship, are unknown in Ghana. This study examines the moderating roles of sex, multimorbidity, and physical activity on the association between neighbourhood social cohesion and functional disability among older people in Ghana. A cross-sectional study of 4,446 people—50 years and older—from WHO's Study on global AGEing and adult health Ghana Wave 2 was employed. Functional disability—WHO Disability Assessment Schedule 2.0—and neighbourhood social cohesion measured with community-level participation, perceived trust and safety were studied. Generalised Logistic regressions with interactional tests were used to examine the associations. A more socially cohesive neighbourhood was significantly associated with a lower functional disability among older people (OR = 0.94, 95%CI: 0.93, 0.94; *P*<0.001). A similar relationship was found for community-level participation (aOR = 0.94, 95%CI: 0.94, 0.95; *P*<0.001) and perceived trust (aOR = 1.00, 95%CI: 0.99, 1.00; *P*<0.001). Community-level participation is associated with a lower functional disability among older people who were physically active (aOR = 0.98, 95%CI: 0.96, 0.99; *P*<0.001). Among the three individual-level measures of neighbourhood social cohesion, only physical activity (OR = 0.98, 95%CI: 0.98, 0.99; *P*<0.01) moderated the association between community-level participation and functional disability. Community-level participation, along with physical activity, may be relevant in improving functional ability among older people. The results highlight the usefulness of policy to ensure a more socially cohesive neighbourhood for older people in Ghana to improve their quality of life.

**Funding:** The author(s) received no specific funding for this work. The funders had no role in study design, data collection and analysis, decision to publish, or preparation of the manuscript.

**Competing interests:** The authors have declared that no competing interests exist.

## Introduction

As the proportion and number of older people grow globally including Ghana [1], opportunities emerge to understand how neighborhood social cohesion can strengthen older people's functional abilities. The term perceived neighborhood social cohesion—defined in this study—refers to a sense of trust, safety, and participation among people who live in the same place [2–4]. Often, neighborhood social cohesion is considered an important element in public health due to its influential role in the wellbeing of older people [5]. As people age, chronic illness and difficulty in daily activities increase. Older people may respond to the decline in physical health by using social resources, including connections, trust, and social bond [4]. The assumption is that when perceptions of trust, safety and participation prevail in a community where older adults are residents, they will be freely involved in all activities, eventually strengthening their functional capacities. Owing to this reasoning and evidence, perceived neighborhood social cohesion may be associated with functional disability—the difficulty individuals experience in engaging in activities of daily living such as bathing, using public transport, caring for households and toileting [6, 7] among older people in Ghana.

Although life expectancy of Ghanaian older people is increasing—52 years in 2005 compared to 58 years in 2019; 2.81% rise [8]—, functional disability among older people in Ghana is noticeable. For instance, a nationally representative study comparing functional disability scores among six countries (China, Mexico, Ghana, South Africa, India, and Russia) that participated in the global study on AGEing and adults' health (SAGE) project reported a lowest in China, highest in India with Ghana and South Africa recording the next higher score [9]. About 90% of older people in Ghana reported a difficulty across domains functioning assessment including cognition, mobility, self-care, getting along with people, engagement in household responsibility and participation in society [9]. A recent qualitative study exploring older people' lived experiences of their functional disability revealed that they feel anxious and restricted in being productive [10]. Given this emerging functional disability prevalence among older people in Ghana, they may rely on informal social networks and other organized events to overcome daily life demands, including dealing with functional disability. In this context, cohesive neighborhoods could be an essential channel to influence functional disability among older people. Though the Ghanaian social structure is mainly communal in several of its social life and social spaces [11], there is very little understanding of the extent to which cohesive neighbourhood is associated with the functional disability of older persons and the nature of the relationship in Ghana. Significant evidence on the relationship between neighbourhood social cohesion and the three measures (perceived safety, trust, and participation) and functional disability exist in western countries. These studies revealed that older people who report higher levels of perceived social cohesion have a lower chance of developing stroke, and myocardial infarction [12, 13]. On the other hand, lower perceived social cohesion through living alone is associated with isolation, reduced social activities and interpersonal ties [4, 14], increased likelihood of developing functional disability [15, 16], thereby perceived social cohesion acting as protective factor against functional disability. In the study by Stephens, Allen [17], a greater accessibility of neighborhoods and higher level of trust among neighbours associate with better mental health.

There is limited research on the relationship between neighborhood social cohesion and functional disability among older people in low- and middle-income countries such as Ghana. Rahman and Singh [18] used data from six countries of WHO SAGE project on functional disability and social cohesion and reported that functional disability is associated with lower social cohesion, with social cohesion being highest among males, rural dwellers, currently married, currently working, better educated, and higher on the socio-economic ladder across all six countries [18]. For example, participating in health behaviour promoting activities such as

physical activity and healthy diet habits may delay the development of functional disability [19, 20]. In Ghana, studies on associations between functional disability and other variables such as food insecurity [21, 22], and social isolation, neighbourhood walkability and loneliness [23, 24], physical activity [25], and long-term care [26] exist with limited understanding of how perceived neighborhood social cohesion and its measures associate with functional disability.

Known factors from western literature that moderate the association between perceived neighborhood social cohesion and functional disability include physical activity [19, 20, 27], chronic conditions such as stroke and heart attack [12] and, age and sex [18, 28]. In Ghana, very little is known about the moderating factors of the association between perceived neighborhood social cohesion and functional disability.

In this study, we examined the potential effect of perceived neighborhood social cohesion on functional disability, particularly identifying how the association between perceived community-level participation, trust, and safety within communities and functional disability are moderated by sex, multi-morbidity, and physical activity in Ghana. The findings will serve as a baseline for policymakers and researchers in Ghana, but also to extract important lessons regarding social policies for ageing populations in low- and middle-income countries.

## Methods

### Study sample

The Study on global AGEing and adult health (SAGE) Ghana Wave 2 data, which was collected within two years (2014–2015) across all regions of Ghana was used in this study. The actual number of participants who participated in SAGE Ghana Wave 2 was 4,704, however only 4,446 responded to questions related to the independent variable—functional disability. SAGE is a longitudinal study that used multistage cluster sampling techniques to explore health and social care information of older adults (50+years) and compared with a few sample of 18-49years across Russia, Mexico, India, China, South Africa and Ghana [29]. In Ghana, Department of Community Health at the University of Ghana with the help of the WHO sort to the implementation of the Wave 2 of SAGE.

### Variables

**Functional disability.** The version 2.0 of the WHO Disability Assessment Schedule (WHODAS) was used to define functional disability in this study. WHODAS 2.0 has a five-response category (none = 0, mild = 1, moderate = 2, severe = 3, and extremely severe = 4). See S1 Appendix for detail. The WHODAS 2.0 is made up of 12-items under six broad areas comprising of participation in society, engaging in life activities, getting along with people, caring for self, moving around and cognition [30] Similar to other studies, we scored the 12 items between 0 and 100 inclusive to determine functional disability severity [9, 31, 32]. We categorised those who scored less than 90.18% as having "no disability" and those with a score of 90.18% or greater as living "with a disability."

### Perceived neighborhood social cohesion

Perceived neighborhood social cohesion was measured from three domains namely perceived community participation, perceived trust, and perceived safety used in WHO SAGE 2 [29]. Nine questions each with a 5-response category were used to measure older people's perception of their participation in their communities (See details at S2 Appendix). In the resulting scale for community-level participation, we found a range from 9 to 45, with higher values denoting higher levels of community participation. The reliability coefficient was 0.87. Three

questions were used to measure perceptions of feeling of trust among participants in the community. Five responses namely to a very great extent = 1, to a great extent = 2, neither great nor small extent = 3, to a small extent = 4, and to a very small extent = 5 (see S2 Appendix). These responses were reversed coded so that a higher number represented higher neighbours trust level. For instance, to a very great extent was coded 5" whereas the to a very small extent was coded 1. The responses were highly reliable at alpha = 0.889. Perceived safety was measured with two variables with 5 response categories with 1 representing completely safe, 2 representing very safe, 3 moderately safe, 4 slightly safe and 5 representing not safe at all. These responses were reversed coded so that a higher number represented higher neighbours perceived feeling of safety. For instance, "completely safe" was coded "5" whereas the "not safe at all "1". The reliability coefficient was 0.847, with values ranging from 2 to 10.

The three individual-level variables 1) Perceived community-level participation, 2) perceived trust and 3) perceived safety were put together using *transformation scale* to measure perceived neighborhood social cohesion. The higher the score the higher levels of perceived neighborhood social cohesion. The values range from 14 to 70 with higher values indicating a more cohesive neighborhood. The Cronbach's alpha of 0.839 represented a high reliability.

**Sex.** The question what your sex is dichotomised as Male = 1 and Female = 2 was used.

**Physical activity (PA).** We measured PA with three separate measures (walking, moderate activity, and vigorous activity). Participants involvement in vigorous activity related work such as *heavy lifting, digging, or chopping wood* categorised as "yes" or "no" were used to measure vigorous activity. *Involvement in work-related moderate activities such as brisk walking, carrying light loads, cleaning, cooking, or washing clothes for at least 10 minutes continuously were used to accessed moderate activity engagement (Yes or No)*. Walk was measured with participants involvement in *walk or use a bicycle (pedal cycle) for at least 10 minutes continuously to get to and from places under yes or no response category*.

The overall PA was generated by aggregating the three measures into "*yes*" representing engagement in one or more of the measures of physical activity) and "no" representing engagement in none of three measures (Cronbach's α = 0.61).

**Multimorbidity.** Multimorbidity was generated from the presence chronic condition: stroke, hypertension, depression, diabetes, angina, arthritis, chronic lung disease, asthma, cataract, and oral health among older people. The response categories measuring the presence of a chronic conditions was "no condition = 1"; "one chronic condition = 2"; and "at least two chronic conditions = 3".

**Covariates.** We considered sociodemographic and health related variables as potential confounders. Sociodemographic included age, marital status (never married = 1, married/cohabiting = 2, separated/divorced = 3, widowed = 4), education (less than primary school = 1, primary education completed = 2, senior high completed = 3, university degree/post = 4), and rural/urban location of residence. Health variable considered in this study was self-reported health status (good = 1, moderate = 2, bad = 3).

## Data analysis

First, we used percentages and means and standard deviations to describe the variables in the study. Second, chi-square, Fisher's test, and t-test to establish relationships between functional disability and neighbourhoods' social cohesion and its domains. Finally, we carried out multivariate logistic regression to estimate the odds ratios (Crudes and adjusted) and 95% confidence intervals (CI) for the associations between perceived neighbourhood social cohesion and functional disability. A moderation analysis (multiplicative terms) of sex, multimorbidity and physical activity in the relationship were estimated at 0.05.

### Ethical consideration

The World Health Organisation Ethical Research Committee provided ethical approval for this study (#ID3925). Participants provided a written and/or verbal consent for the study [29].

## Results

### Characteristics of study participants

S1 Table shows the features of study participants. The average age of participants with functional disability was approximately 74 years, with higher proportion of females reporting higher functional disability compared to the proportion of males (64.5% vs 35.5%). A high prevalence of functional disability was found among widowed (46.5%), rural dwellers (61.1%), senior high school leavers (39.5%), those who reported health as bad (68.2%) and lived with at least two chronic conditions (45.0%). A high prevalence of older people who reported absence physical activity engagement of any kind suffered functional disability (55.6%) compared to those who are physically active (55.6% vs 44.4%). Older people with no functional disability had a higher mean perceived community level participation score compared with older people with functional disability (24.8 vs. 18.3, P<0.001).

### Neighbourhood social cohesion and functional disability

In the unadjusted model, overall perceived neighbourhood social cohesion was statistically associated with functional disability (OR = 0.94, 95%CI: 0.93, 0.94; *P*<0.001). When adjusted for potential confounders including age, sex and marital status, the strength of association existing between overall perceived neighbourhood social cohesion and functional disability was still significant and of similar magnitude (aOR = 0.94, 95%CI: 0.93, 0.95; *P*<0.001) (S2 Table).

In S3 Table, the associations between perceived community-level participation level and perceived trust with functional disability were significant after adjusting for potential confounders. However, the adjusted association between perceived safety and functional disability (after controlling for potential confounders) shows to be statistically insignificant.

S4 Table shows the interactional effect of sex, multimorbidity and physical activity on the neighborhood association with functional disability. None of the interactional variables studied moderated the association between overall neighborhood social cohesion and functional disability.

Among the three measures of perceived neighborhood social cohesion studied, the association between perceived community-level participation and functional disability was significant with the moderators (OR = 0.94, 95%CI: 0.94, 0.95; *P*<0.001). Even among the three moderators, only physical activity moderated the association between perceived community-level participation and functional disability. Older people participating in community and physical activities were 2% less likely to experience functional disability (OR, 0.98, 95%CI: 0.98, 0.99; *P*<0.01) (see S5 Table).

## Discussion

Evidence on the effect of sex, physical activity, and multi-mobility on the association between perceived social cohesion and functional disability is least established in the gerontological literature from low- and middle- income countries. To contribute to addressing this knowledge gap, the aim of this study was to determine the moderating role of sex, physical activity, and multi-morbidity on the association between perceived neighbourhood social cohesion and functional disability. The important findings for policy and practice implications are discussed.

The current study finding that a more socially cohesive neighbourhood was associated with a lower functional disability among older people confirms available studies in other low- and middle-income countries [33–35]. Adding to this evidence, in a high income country such as Japan, previous gerontological study has reported that social cohesive neighbourhoods reduce functional disability [35]. In a related study, Aida, Kondo [34] reported that higher incidence of functional disability is linked to lower community social capital among women in Japan. Three important reasons may explain the relationship between social cohesion and functional disability among older people in the literature. Firstly, with high socially cohesive neighborhoods, older people may have higher odds of getting access to social support when they have health problems [36]. With access to social support, they can seek early health treatment(s) to prevent the onset of functional disability [16]. Secondly, social cohesion reduces the development of functional disability through social networking and group activities which result in positive health behaviour such as physical activity and healthy diets [37, 38]. Lastly, increased social cohesion is linked to improved mental and physical wellbeing which lessen the functional disability [16, 39]. Our results thus suggest that older people with higher socially cohesive neighbourhoods tend to demonstrate better functional and psychological health. Given the significant effect of social cohesion on functional disability, ensuring socially cohesive neighborhood is likely to improve functional status of older people and also reduce the risk of functional disability associated with ageing [16]. This implies that to ensure improved health of older people, social cohesion such as social connections and trust should be considered as a health priority [35]. Our results further imply that to lessen functional disability, community-based measures which foster social capital may be important [34].

It is important to highlight that perceived trust and perceived community-level participation, which are part of the framework for measuring socially cohesive neighborhoods were both associated with functional disability. Interestingly, the study revealed that increased in social trust and community-level participation reduces functional disability. This finding underscores the need to promote social trust and community participation in old age to reduce the risks of development of functional disability at the community levels. Clear evidenced based policy initiatives are required to be implemented to foster social trust building at the community level. The findings from this study are consistent with the observations made by previous gerontological studies conducted elsewhere [40, 41]. Corroborating the present findings, a study conducted in China reiterates how increase in social participation in old age predicts lower risk of the developing functional disability [42]. This finding further affirms Fujihara, Miyaguni [41] assertion that older people with increased level of community participation (such as sports) are less likely to report better functional health. This result further reinforces Chen, Min [40] finding that participating in community moderates functional disability and life satisfaction association.

Our finding that increased level of community-level participation lowers the risk of functional disability in old age may be attributed to four possible reasons. In the first place, older people who participate actively and more in community events such as communal labour and other outdoor activities are less likely to be sedentary and as a result have better functional health. Secondly, older people with increased community-level participation have lower odds of being homebound compared with those with low level of community participation thereby reducing their financial disability [41]. Thirdly, social participation enhances access to health-relevant information in old age, which is important to promoting functional health. Lastly, social participation enables older people to stay active (such as dressing each day to leave home) and these daily functions help to improve their functional health [43].

The above reasons are based on our finding that physical activity moderated the association between community-level participation and functional disability. These findings highlight

several important policy implications. First, preventive programme and/or policy to improve functional ability among older people should encourage older people to participate frequently and more actively in social events at the community levels. Second, healthcare providers rendering care to older people need to gain a better understanding of the relevance of socially cohesive neighborhoods in improving functional ability in old age. Third, to improve functional ability of older people, diverse indicators for measuring socially cohesive neighborhood (such as social capital) should be considered [39]. This is because several factors such as trust and community-level participation as a dimension of socially cohesive neighbourhoods have proven to reduce the risk of functional disability in old age. Lastly, social workers can use the media to promote the need for physical activity in old age for a desired functional health and quality of life. Social workers working within communities can sensitize older adults and families on the need to incorporate physical activity in their daily activities.

The study has some limitations that need to be considered. The extent of the analysis is limited by the availability of data. Issues regarding internal validity—e.g., due to other variables that could be mediating the relationship between social cohesion and functional disability—and external validity that need to be considered. However, results are in line with other studies showing a promising research and policy area that has not been extensively explored, especially in Ghana and other low- and middle-income contexts. Data from the SAGE study Wave 2 collected during 2014/2015 could be dated; however, given the current trends in population ageing and functional disability, it is expected that results have even more relevance today.

Our findings arise as important for policymakers, since they highlight the relevance of social policies that help building social cohesion not just as important per se and to improve community wellbeing but also as a strategy to address the expected increase in long-term care needs coming from population ageing and the rise in the prevalence of functional disability. The results are important for Ghana and other low- and middle-income countries since can be seen as an efficient policy—freeing two birds with one key—for addressing the pressing social security demands in these countries.

## Conclusion

Findings from this nationally representative study demonstrated the importance of a socially cohesive neighbourhood in reducing the risk of functional disability among older people, through physical activity in their long-term care. The findings have implications for policy makers to ensure social cohesiveness is improved by fostering the establishment of social support groups and local community network groups in any health and social care systems that seek to address an aspect of the long-term care needs of older people. Setting up community centres where older people could enhance companionship and engage in physical health enhancing activities such as exercises and walks with peers would be essential. Such facility will offer older people the privilege for social interaction to reduce incidence of loneliness and its associated functional disability experienced by older people. Healthcare providers should also emphasise on the need for older people to be physically active while family friendly relationships should be strengthened. A holistic approach is needed to ensure a socially cohesiveness community rather than a single entity. Further study is warranted to establish the nature and trajectory of community-level participation that help reduce functional disability.

## Supporting information

**S1 Checklist. STROBE statement—checklist of items that should be included in reports of observational studies.**
(DOCX)

**S1 Table. Univariate and bivariate analysis of independent variables and functional disability.**
(DOCX)

**S2 Table. Relationship between perceived neighbourhood social cohesion (overall) and functional disability adjusted for confounders.**
(DOCX)

**S3 Table. Effects of confounding on the relationship between measures of perceived neighbourhood social cohesion and functional disability.**
(DOCX)

**S4 Table. Sex, multimorbidity and physical activity moderation on perceived neighbourhood social cohesion association with functional disability.**
(DOCX)

**S5 Table. Sex, multimorbidity and physical activity moderation on perceived community-level participation association with functional disability.**
(DOCX)

**S1 Appendix. List of the 12 variables included in the WHODAS score and cut points.**
(DOCX)

**S2 Appendix. Perceived neighborhood social cohesion measure.**
(DOCX)

**S1 Data.**
(XLSX)

## Author Contributions

**Conceptualization:** Kofi Awuviry-Newton, Dinah Amoah.

**Data curation:** Paul Kowal.

**Formal analysis:** Kofi Awuviry-Newton.

**Investigation:** Paul Kowal.

**Methodology:** Kofi Awuviry-Newton, Dinah Amoah.

**Project administration:** Paul Kowal.

**Software:** Kofi Awuviry-Newton.

**Supervision:** Kofi Awuviry-Newton, Daniel Doh, Kwadwo Ofori-Dua, Paul Kowal.

**Validation:** Kofi Awuviry-Newton, Dinah Amoah, Daniel Doh, Seth Christopher Yaw Appiah.

**Visualization:** Kofi Awuviry-Newton, Daniel Doh, Kwadwo Ofori-Dua.

**Writing – original draft:** Kofi Awuviry-Newton, Dinah Amoah, Daniel Doh, Williams Agyemang-Duah, Kwadwo Ofori-Dua, Paul Kowal, Seth Christopher Yaw Appiah.

**Writing – review & editing:** Kofi Awuviry-Newton, Dinah Amoah, Daniel Doh, Williams Agyemang-Duah, Kwadwo Ofori-Dua, Paul Kowal, Seth Christopher Yaw Appiah.

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
