## [Editor Report · Decision Letter 0]

24 May 2023

PONE-D-23-14414Perceived neighborhood social cohesion and functional disability among older adults in Ghana: the moderating roles of sex, physical activity, and multi-morbidityPLOS ONE

Dear Dr. Awuviry-Newton,

Thank you for submitting your manuscript to PLOS ONE. After careful consideration, we feel that it has merit but does not fully meet PLOS ONE’s publication criteria as it currently stands. Therefore, we invite you to submit a revised version of the manuscript that addresses the points raised during the review process.

We look forward to receiving your revised manuscript.

Kind regards,

Nestor Asiamah, PhD

Academic Editor

PLOS ONE

Journal Requirements:

Additional Editor Comments:

Dear authors,

Thanks for submitting your manuscript to PLOS ONE. Your manuscript's similarity is currently at 34% (without the reference list and author information), which is above the minimum value allowed by PLOS ONE. Ideally, your manuscript should be 10% or less. Can I ask you to edit your work carefully to bring its similarity to the acceptable level? The file showing your similarity index is attached. Please note that your manuscript has not been peer-reviewed, so this decision is only formative and applies to the manuscript's originality.

I look forward to recieving the edited manuscript.

---

## [Author Response · Author response to Decision Letter 0]

12 Jul 2023

Response to reviewers

Dear editor and reviewers, 

We appreciate the privilege extended to us to revise the manuscript. We acknowledge that the comments raised are relevant and they have substantially improved the manuscript. 

Response

We have formatted the manuscript to conform to the Plos One’s style requirements. 

Response

This was an error and so we have rectified this error. The funding declaration now reads as “The authors received no specific funding for this work.”

Response

Authors have provided the minimal dataset underlying this manuscript. We have uploaded this as a supplementary document. 

4. Please include your full ethics statement in the ‘Methods’ section of your manuscript file. In your statement, please include the full name of the IRB or ethics committee who approved or waived your study, as well as whether you obtained informed written or verbal consent. If consent was waived for your study, please include this information in your statement as well. 

Response

We have provided the ethics statement in the “Methods” section, with a mention of the ethics committee and reflection on participants consent. It now states as “The World Health Organisation Ethical Research Committee provided ethical approval for this study (#ID3925). Participants provided a written and/or verbal consent for the study” under “ethical consideration”. 

5. Please review your reference list to ensure that it is complete and correct. If you have cited papers that have been retracted, please include the rationale for doing so in the manuscript text or remove these references and replace them with relevant current references. Any changes to the reference list should be mentioned in the rebuttal letter that accompanies your revised manuscript. If you need to cite a retracted article, indicate the article’s retracted status in the References list and also include a citation and full reference for the retraction notice.

Response 

All references have been cross-checked. All citations have their corresponding list of references towards the end of the manuscript. 

Response

Captions for all supporting documents have been included at the end of the manuscript. 

7. Thanks for submitting your manuscript to PLOS ONE. Your manuscript's similarity is currently at 34% (without the reference list and author information), which is above the minimum value allowed by PLOS ONE. Ideally, your manuscript should be 10% or less. Can I ask you to edit your work carefully to bring its similarity to the acceptable level? The file showing your similarity index is attached. Please note that your manuscript has not been peer-reviewed, so this decision is only formative and applies to the manuscript's originality.

Response 

I have paraphrased all areas. This are evident on the manuscript.

---

## [Decision Letter · Decision Letter 1]

11 Sep 2023

PONE-D-23-14414R1Perceived neighborhood social cohesion and functional disability among older adults in Ghana: the moderating roles of sex, physical activity, and multi-morbidityPLOS ONE

Dear Dr. Awuviry-Newton,

Thank you for submitting your manuscript to PLOS ONE. After careful consideration, we feel that it has merit but does not fully meet PLOS ONE’s publication criteria as it currently stands. Therefore, we invite you to submit a revised version of the manuscript that addresses the points raised during the review process.

Though your manuscript has some merit, it needs some improvement before it can be published. Please revise your manuscript based on the comments of the two reviewers. 

We look forward to receiving your revised manuscript.

Kind regards,

Nestor Asiamah, PhD

Academic Editor

PLOS ONE

Additional Editor Comments:

Please revise the manuscrip based on the comments of the two reviewers.

Reviewers' comments:

Reviewer's Responses to Questions

**Comments to the Author**

1. If the authors have adequately addressed your comments raised in a previous round of review and you feel that this manuscript is now acceptable for publication, you may indicate that here to bypass the “Comments to the Author” section, enter your conflict of interest statement in the “Confidential to Editor” section, and submit your "Accept" recommendation.

Reviewer #1: (No Response)

Reviewer #2: (No Response)

2. Is the manuscript technically sound, and do the data support the conclusions?

Reviewer #1: Yes

Reviewer #2: Partly

3. Has the statistical analysis been performed appropriately and rigorously? 

Reviewer #1: Yes

Reviewer #2: No

4. Have the authors made all data underlying the findings in their manuscript fully available?

Reviewer #1: Yes

Reviewer #2: Yes

5. Is the manuscript presented in an intelligible fashion and written in standard English?

Reviewer #1: Yes

Reviewer #2: Yes

6. Review Comments to the Author

Reviewer #1: 1. This is a well written article, and I really enjoyed reading it. It will be a worthwhile addition to the journal. The authors' compelling Introduction argues for the necessity of their study. In general, the methods used were clearly described. The authors' presentation of the findings is engaging and convincing. The results were integrated into the more general literature during the discussion. Findings and recommendations were explicit. However, the authors could improve the paper by considering some minor points/ suggestions which can be looked at.

2. Page 2. Introduction: line 6. It seems there is omission of a word in: “due to its influential role on the wellbeing older people”

3. Methods - Sample size: Page 5. Line 1 Authors did not provide any justification for the use of the instrument and also

did not provide evidence of the regions where the sample size (4406) were drawn from in Ghana.

4. Page7, line 12; It is not clear why “pedal cycle” was categorized as walk. The question is what type of walk? Since the energy expenditure in both activities are not the same unless walk is properly defined.

5. page 9-10 (Tables). Authors did not provide any Table /link to how PA was measured

6. Page 10 Tables. Authors should be consistence with the use of sex and gender. Though sex was used throughout the work, gender was used in the tables instead

7. Page 13. Paragraph 1 line 11. The last sentence. “Lastly there is the need to promote PA……”

It will be appropriate if authors provide measures or strategies that can be adopted to promote older people’s participation in PA. authors should also provide the agency or body to which this recommendation is been address to.

Reviewer #2: General comments

There were no line numbers in the document. Thus, it made it quite difficult to make my references to identified issues. The manuscript needs to undergo grammar checking and sentence construction.

Introduction

1. “Given this significant functional disability prevalence among older people in Ghana,”

The use of the word “significant” in this sentence is not justified, given the context in which it was used. A much milder term may be used.

2. “the extent to which cohesive neighbourhood associate with functional disability of older persons and the nature of the relationship, is unexplored in Ghana”

The use of the word “unexplored” should be revised. There are works relating to the subject under investigation.

3. In text citation number 17, should be written properly.

4. In low-and middle-income countries including Ghana

A comma (,) should be placed after Ghana

5. In low-and middle-income countries including Ghana evidence on neighbourhood social cohesion and its measures, and functional disability among older people is scant.

Again, much milder constructions should be considered. There are works in this area.

Methods

Physical activity

1. “the overall PA”

Start with a capital T

Results

1. The significance level values were not stated in the writing of the results. E.g., (OR=0.94, 95%CI: 0.93, 0.94). Only the odds ratio and confidence intervals were stated. Authors can add the P values. Also, I am aware the adjusted odds ratios are written as “AOR”. Authors should kindly check in on this. These should be updated in the abstract results also.

2. That is, participating in in community activities and report engaging in physical activity by older adults were 2% less likely to experience functional disability (OR, 0.98, CI: 0.98, 0.99).

There are typos in the sentence. Also, I do not seem to follow the 2% less likelihood from the stated results (OR, 0.98, CI: 0.98, 0.99).

3. Kindly check the reporting of your regression results again.

Discussion

1. The current study finding that a more perceived socially cohesive neighbourhood was associated with a lower functional disability among older people confirms available studies in other low- and middle-income countries [33-35].

There are typos in the sentence. Also, Reference number 34 says something about Japanese. I do not think Japan is a low- and middle-income country. Please check your reference.

7. PLOS authors have the option to publish the peer review history of their article (what does this mean?). If published, this will include your full peer review and any attached files.

Reviewer #1: No

Reviewer #2: No

---

## [Author Response · Author response to Decision Letter 1]

16 Sep 2023

Responses to Author

Reviewer 1 

 Reviewer Comments Our Response

 1. This is a well written article, and I really enjoyed reading it. It will be a worthwhile addition to the journal. The authors' compelling Introduction argues for the necessity of their study. In general, the methods used were clearly described. The authors' presentation of the findings is engaging and convincing. The results were integrated into the more general literature during the discussion. Findings and recommendations were explicit. However, the authors could improve the paper by considering some minor points/ suggestions which can be looked at. We appreciate this complement. Authors have responded to all comments. 

 2. Page 2. Introduction: line 6. It seems there is omission of a word in: “due to its influential role on the wellbeing older people” This sentence has been corrected. 

 3. Methods - Sample size: Page 5. Line 1 Authors did not provide any justification for the use of the instrument and also

did not provide evidence of the regions where the sample size (4406) were drawn from in Ghana. The sentence “The Study on global AGEing and adult health (SAGE) Ghana Wave 2 data, which was collected within two years (2014-2015) across all regions of Ghana was used in this study” has been provided indicating that it was a countrywide study. The details and reason for adopting multistage cluster sampling has been explained by the paper references. 

 4. Page7, line 12; It is not clear why “pedal cycle” was categorized as walk. The question is what type of walk? Since the energy expenditure in both activities are not the same unless walk is properly defined. It is true that the energy expenditure from these two activities are not the same, however, in this study we classified walk to represent any participant who reported taking part of any of this activities. Therefore, we interpreted the findings in relation to this measurement. 

 5. page 9-10 (Tables). Authors did not provide any Table /link to how PA was measured We explained under “Physical activity” how PA was measured. PA was measured with walking, moderate activity, vigorous activity. 

 6. Page 10 Tables. Authors should be consistence with the use of sex and gender. Though sex was used throughout the work, gender was used in the tables instead We have replaced “gender” with “sex” throughout. 

 7. Page 13. Paragraph 1 line 11. The last sentence. “Lastly there is the need to promote PA……”

It will be appropriate if authors provide measures or strategies that can be adopted to promote older people’s participation in PA. authors should also provide the agency or body to which this recommendation is been address to.

 The sentence now reads “Lastly, social workers can use the media to promote the need for physical activity in old age for a desired functional health and quality of life. Social workers working within communities can sensitize older adults and families on the need to incorporate physical activity in their daily activities”.

Reviewer 2 

 There were no line numbers in the document. Thus, it made it quite difficult to make my references to identified issues. We have included line numbers in the manuscript.

 The manuscript needs to undergo grammar checking and sentence construction. We have done a thorough editing of the manuscript and checked for grammar as much as possible.

Introduction 1. “Given this significant functional disability prevalence among older people in Ghana,”

The use of the word “significant” in this sentence is not justified, given the context in which it was used. A much milder term may be used.

 We have revised the sentence as: 

Given this emerging functional disability prevalence among older people in Ghana, they may rely on informal social networks and other organized events to overcome daily life demands, including dealing with functional disability.

 2. “the extent to which cohesive neighbourhood associate with functional disability of older persons and the nature of the relationship, is unexplored in Ghana”

The use of the word “unexplored” should be revised. There are works relating to the subject under investigation. 

 We have revised the sentence to:

There is very little understanding of the extent to which cohesive neighbourhood is associated with the functional disability of older persons and the nature of the relationship in Ghana.

 3. In text citation number 17, should be written properly. corrected

 4. In low-and middle-income countries including Ghana

A comma (,) should be placed after Ghana Corrected. 

 5. In low-and middle-income countries including Ghana evidence on neighbourhood social cohesion and its measures, and functional disability among older people is scant.

Again, much milder constructions should be considered. There are works in this area. 

 The sentence has been reworded:

There is limited research on the relationship between neighborhood social cohesion and functional disability among older people in low- and middle-income countries such as Ghana.

Method 1. “the overall PA”

Start with a capital T

 Corrected – See line 182

Results 1. The significance level values were not stated in the writing of the results. E.g., (OR=0.94, 95%CI: 0.93, 0.94). Only the odds ratio and confidence intervals were stated. Authors can add the P values. Also, I am aware the adjusted odds ratios are written as “AOR”. Authors should kindly check in on this. These should be updated in the abstract results also.

 P values have been provided at respective places. All adjusted odds ratios have been written us “aOR”

 2. That is, participating in in community activities and report engaging in physical activity by older adults were 2% less likely to experience functional disability (OR, 0.98, CI: 0.98, 0.99).

There are typos in the sentence. Also, I do not seem to follow the 2% less likelihood from the stated results (OR, 0.98, CI: 0.98, 0.99).

 The sentence has been reworded:

Even among the three moderators, only physical activity moderated the association between perceived community-level participation and functional disability. Older people participating in community and physical activities were 2% less likely to experience functional disability (OR, 0.98, CI: 0.98, 0.99) (see S5 Table).

 3. Kindly check the reporting of your regression results again.

 We have checked and can confirm that is correct. 

Discussion 1. The current study finding that a more perceived socially cohesive neighbourhood was associated with a lower functional disability among older people confirms available studies in other low- and middle-income countries [33-35].

There are typos in the sentence. Also, Reference number 34 says something about Japanese. I do not think Japan is a low- and middle-income country. Please check your reference.

 The sentence has been corrected

---

## [Decision Letter · Decision Letter 2]

4 Oct 2023

Perceived neighborhood social cohesion and functional disability among older adults in Ghana: the moderating roles of sex, physical activity, and multi-morbidity

PONE-D-23-14414R2

Dear Dr. Awuviry-Newton,

We’re pleased to inform you that your manuscript has been judged scientifically suitable for publication and will be formally accepted for publication once it meets all outstanding technical requirements.

Kind regards,

Nestor Asiamah, PhD

Academic Editor

PLOS ONE

Additional Editor Comments (optional):

Congratulations to the authors!

Reviewers' comments:

Reviewer's Responses to Questions

**Comments to the Author**

1. If the authors have adequately addressed your comments raised in a previous round of review and you feel that this manuscript is now acceptable for publication, you may indicate that here to bypass the “Comments to the Author” section, enter your conflict of interest statement in the “Confidential to Editor” section, and submit your "Accept" recommendation.

Reviewer #1: All comments have been addressed

Reviewer #2: All comments have been addressed

2. Is the manuscript technically sound, and do the data support the conclusions?

Reviewer #1: Yes

Reviewer #2: Yes

3. Has the statistical analysis been performed appropriately and rigorously? 

Reviewer #1: Yes

Reviewer #2: Yes

4. Have the authors made all data underlying the findings in their manuscript fully available?

Reviewer #1: Yes

Reviewer #2: Yes

5. Is the manuscript presented in an intelligible fashion and written in standard English?

Reviewer #1: Yes

Reviewer #2: Yes

6. Review Comments to the Author

Reviewer #1: All my comments and suggestions from the initial peer review were properly addressed by the authors.

Reviewer #2: Introduction

The introduction now reads better. For line 105 to 106 which reads “The findings will serve as a baseline for policymakers and researchers in Ghana, but also to extract important lessons regarding social policies for ageing populations in low- and middle-income countries”. I do not think these findings will serve as baseline study, the sentence can read “The findings will contribute to important lessons regarding social policies for ageing populations in low- and middle-income countries”.

Methods

This section is sound.

Results

All comments have been addressed except Adjusted odd ratios are still written as “aOR”.

Discussion

The section is sound.

7. PLOS authors have the option to publish the peer review history of their article (what does this mean?). If published, this will include your full peer review and any attached files.

Reviewer #1: No

Reviewer #2: No

---

## [Editor Report · Acceptance letter]

22 Jan 2024

PONE-D-23-14414R2 

PLOS ONE

Dear Dr. Awuviry-Newton, 

I'm pleased to inform you that your manuscript has been deemed suitable for publication in PLOS ONE. Congratulations! Your manuscript is now being handed over to our production team.

Kind regards, 

on behalf of

Dr. Nestor Asiamah 

Academic Editor

PLOS ONE